Seasonal and longitudinal water quality dynamics in three effluent-dependent rivers in Arizona

Hamdhani Hamdhani 1 2 hamdhani@arizona.edu
Eppehimer Drew E. 2
Quanrud David M. 2
Bogan Michael T. 2
1 Department of Aquatic Resources Management, Mulawarman University , Samarinda, East Kalimantan , Indonesia
2 School of Natural Resources and the Environment, University of Arizona , Tucson, AZ , United States
Riaz Muhammad
Electronic publication date: 2023 Mar 29
Publication date: 2023
Volume: 11
Electronic Location ID: e15069
Received 2022 Aug 19; Accepted 2023 Feb 23
Copyright: © 2023 Hamdhani et al.
Copyright year: 2023
Copyright holder: Hamdhani et al.
License: This is an open access article distributed under the terms of the Creative Commons Attribution License, which permits unrestricted use, distribution, reproduction and adaptation in any medium and for any purpose provided that it is properly attributed. For attribution, the original author(s), title, publication source (PeerJ) and either DOI or URL of the article must be cited.
License URL: https://creativecommons.org/licenses/by/4.0/

Keywords: Wastewater, In-stream natural purification, Wastewater treatment plant, Aquatic organism, Urban arid region

Funding: Indonesia Endowment Funds for Education University of Arizona Lincoln Institute’s Babbitt Dissertation Fellowship Program During the sampling and writing of this study, Michael T Bogan was supported by start-up funding from the University of Arizona and Drew Eppehimer was supported by the Lincoln Institute’s Babbitt Dissertation Fellowship Program. The funders had no role in study design, data collection and analysis, decision to publish, or preparation of the manuscript.

==============================
Effluent-fed streams, which receive inputs from wastewater treatment plants, are becoming increasingly common across the globe as urbanization intensifies. In semi-arid and arid regions, where many natural streams have dried up due to over extraction of water, many streams rely completely on treated effluent to sustain baseflow during dry seasons. These systems are often thought of as ‘second-class’ or highly disturbed stream ecosystems, but they have the potential to serve as refuges for native aquatic biota if water quality is high, especially in areas where few natural habitats remain. In this study, we investigated seasonal and longitudinal water quality dynamics at multiple sites across six reaches of three effluent-dependent rivers in Arizona (USA) with the objective (1) to quantify changes in effluent water quality due to distance traveled and season/climate and (2) to qualify whether water quality conditions in these systems are sufficient to support native aquatic species. Study reaches ranged in length from 3 to 31 km and in geographic setting from low desert to montane conifer forest. We observed the lowest water quality conditions (e.g., elevated temperature and low dissolved oxygen) during the summer in low desert reaches, and significantly greater natural remediation of water quality in longer vs. shorter reaches for several factors, including temperature, dissolved oxygen and ammonia. Nearly all sites met or exceeded water quality conditions needed to support robust assemblages of native species across multiple seasons. However, our results also indicated that temperature (max 34.2 °C), oxygen levels (min 2.7 mg/L) and ammonia concentrations (max 5.36 mg/L N) may occasionally be stressful for sensitive taxa at sites closest to effluent outfalls. Water quality conditions may be a concern during the summer. Overall, effluent-dependent streams have the capacity to serve as refuges for native biota in Arizona, and they may become the only aquatic habitat available in many urbanizing arid and semi-arid regions.

Introduction

In recent years, various studies from across the globe have reported overuse of surface and groundwater resources (Gleeson et al., 2012; Voss et al., 2013; Schewe et al., 2014; Wada & Bierkens, 2014). As a consequence, many rivers and streams have experienced flow regime alterations, including the loss of perennial flow and prolonged dry periods, especially in arid regions (Logan, 2006; Goodrich et al., 2018). Concurrent with this depletion of natural water resources, wastewater treatment plants have been constructed in many urban and suburban areas, and effluent from these plants is usually discharged into adjacent streams after treatment (Tchobanoglous, Burton & Stensel, 2003). These effluent inputs are especially noticeable in streams that have dried up due to water extraction, and now depend entirely on effluent to sustain baseflow during dry seasons (Luthy et al., 2015). Such streams are known as being effluent-dependent (100% effluent during baseflow: Du et al., 2015) and are increasingly common across the globe (Hamdhani, Eppehimer & Bogan, 2020).

In many regions, the discharge of effluent into dewatered streams incidentally helps re-create aquatic and riparian habitat that was previously lost (Boyle & Fraleigh, 2003; Luthy et al., 2015; Wolfand et al., 2022), and can even support endangered aquatic and riparian species. For example, endangered California red-legged frogs (Rana draytonii) and San Francisco garter snakes (Thamnophis sirtalis tetrataenia) are found in effluent-dependent streams in California (Halaburka et al., 2013; Luthy et al., 2015) and endangered Gila topminnow (Poeciliopsis occidentalis) are found in similar streams in Arizona (Minckley, 1999; Sonoran Institute, 2017). However, the quality of wastewater treatment processes may limit the potential uses of effluent for ecosystem restoration in other cases (Canobbio et al., 2009; Matamoros & Rodríguez, 2017; Hamdhani, Eppehimer & Bogan, 2020).

Water quality in effluent-dependent streams can be quite different from that in natural streams (Brooks, Riley & Taylor, 2006; Hamdhani, Eppehimer & Bogan, 2020; Nikel et al., 2021). Differences noted in effluent-dominated and effluent-dependent streams include elevated values for water temperatures (Boyle & Fraleigh, 2003; Canobbio et al., 2009) and nutrient levels, such as nitrate (Hur et al., 2007; Chen et al., 2009), ammonium/ammonia (Gafny, Goren & Gasith, 2000; Bergbusch et al., 2021), and phosphate (Birge et al., 1989; Chen et al., 2009). Additionally, lower dissolved oxygen levels are frequently observed in stream reaches closest to effluent outfalls (Birge et al., 1989; Matamoros & Rodríguez, 2017).

However, as effluent flows through a stream channel, natural remediation processes (e.g., aeration, nutrient uptake) can occur and lead to improved water quality conditions (Boyle & Fraleigh, 2003). Previous studies demonstrated that longitudinal distance from the effluent source is an important factor in these processes (Growns et al., 2009; Hughes et al., 2014), which suggests that the degree of remediation might differ depending on the length of effluent-dependent reaches. Additionally, studies from natural stream systems demonstrate that seasonality and climate also play important roles in influencing water quality (e.g., Pionke et al., 1999; Morrill, Bales & Conklin, 2005), so these factors likely are important in effluent-dependent streams as well. However, few studies have examined how water quality changes with varying longitudinal distances from effluent inputs and how those changes may differ by season or climate zone.

In this study, we examined the seasonal dynamics of basic water quality parameters (e.g., temperature, dissolved oxygen, pH, nutrients) in six effluent-dependent reaches of differing lengths in three river systems across a climate gradient in Arizona (USA). Our goals were to investigate: (1) how water quality changes with distance from effluent outfalls, (2) how water quality and longitudinal patterns change with season and climate, and (3) whether water quality conditions in these systems are sufficient to support native aquatic species. We hypothesized that: (1) water quality would improve with increasing distance downstream from effluent outfalls due to natural remediation processes, (2) water quality conditions would deteriorate in summer, especially in reaches with hotter climates, and (3) water quality conditions would be sufficient to provide suitable habitat for native fishes and other aquatic organisms adapted to life in natural streams of the region.

Methodology

Site description

We sampled water quality across six effluent-dependent river reaches in Arizona: three reaches in the Santa Cruz River basin, two reaches in the Rio de Flag basin, and one reach in the Salt River basin (Fig. 1). Groundwater levels are at least 40 m below the streambed surface for all six reaches, so each reach would dry completely during baseflow if effluent was not being added to the channel (Carlson et al., 2011; Arizona Department of Environmental Quality, n.d). Additionally, all reaches with the exception of the Rio de Flag, fall within Active Management Areas of the Arizona Department of Water Resources, and groundwater withdrawals are monitored to achieve safe yield. Study reach lengths ranged from 3 to >24 km. The reach in the Salt River and both reaches of the Rio de Flag were short (~3–5 km). In the Santa Cruz River, one reach was medium in length (~9 km) and two others were long (>24 km) (Table 1). Downstream of all six study reaches, the channels are dry except during periods of flooding.

Figure 1 Map of the study area showing effluent-dependent stream reaches, effluent outfalls, and sampling sites in the state of Arizona.

Each reach has been magnified to show more detail and each inset map has a corresponding scale bar.

Table 1 Sampling locations by river and reach, with information about the length of the reach, the number of sites sampled within each reach, and the number of monthly and seasonal sampling events for each reach.

River	Reach	Length (km)	# Sites sampled	# Months sampled	Sampling events by season	
Santa Cruz River	Tres Rios	31	8	12	Winter (3x)

Spring (3x)

Summer (3x)

Fall (3x)

	
Nogales Int.	24	5	6	Winter (2x)

Spring (1x)

Summer (2x)

Fall (1x)

	
Agua Nueva	9	3	4	Winter (1x)

Spring (1x)

Summer (1x)

Fall (1x)

	
Salt River	91st Ave	6	3	4	Winter (1x)

Spring (1x)

Summer (1x)

Fall (1x)

	
Rio de Flag	Wildcat Hill	4	4	4	Winter (1x)

Spring (1x)

Summer (1x)

Fall (1x)

	
Central	3	3	4	Winter (1x)

Spring (1x)

Summer (1x)

Fall (1x)

	

The Santa Cruz River basin (22,000 km2) is located in southern Arizona and northern Mexico (Webb et al., 2014). The climate is characterized by hot summers (average high July temp: 38 °C) and moderately cool winters (average low January temp: 5 °C). Annual average precipitation is ~300 mm and rainfall is bimodal, with the months of January and August being the wettest and May and November being the driest (data provided by NOAA ESRL Global Monitoring Division, Boulder, Colorado, USA (http://esrl.noaa.gov/)). The uppermost study reach in the Santa Cruz Basin (~25 km long) is supported by the Nogales International Wastewater Treatment Plant. This plant was constructed in 1943, upgraded in 2009, and discharges approximately 57 million liters of tertiary-treated effluent into the river each day (Santa Cruz County, 2019). The second Santa Cruz reach is shorter (~9 km) and is supported by the Agua Nueva Water Reclamation Facility (WRF), a relatively new facility completed in 2014 that replaced the Roger Road Wastewater Treatment Plant constructed in 1951, and discharges approximately 30 million liters of tertiary-treated effluent into the river each day (Sonoran Institute, 2017). The third and longest (~31 km) reach is supported by the Tres Rios WRF, which was built in 1978, upgraded and modernized in 2014, and discharges approximately 115 million liters of tertiary-treated effluent each day (Sonoran Institute, 2017). In addition to baseflow from the treatment plants, all three study reaches in the Santa Cruz River experience seasonal flood events from precipitation runoff, which can surpass 280 m3/s. The three perennial study reaches are separated by dry reaches (Fig. 1) and are only briefly connected during large flood events.

The Salt River basin (35,000 km2) is located in central Arizona, with its headwaters in the White Mountains of eastern Arizona. The study reach is characterized by hot summers (average high July temp: 41 °C) and mild winters (average low December temp: 7 °C). Annual average precipitation is ~204 mm, and rainfall is bimodal. The months of March and July are the wettest, and June and October the driest (data provided by NOAA ESRL Global Monitoring Division, Boulder, Colorado, USA (http://esrl.noaa.gov/)). Our study reach (~6 km long) is supported by the 91st Avenue Wastewater Treatment Plant near Phoenix, which began discharging into the river in 1958, was upgraded in 2011, and processes an average of 530 million liters of tertiary treated effluent per day (US EPA, 2016).

The Rio de Flag basin (518 km2) is located in the conifer forest-dominated, high elevation mountains (>2,000 m) of north-central Arizona (Figs. 1 and 2). The climate is characterized by moderate to warm temperatures in summer (average high July temp: 27 °C) and cold winters (average low January temp: −12 °C). Annual average precipitation is ~588 mm, and rainfall is bimodal. The months of February and August are wettest, and June and November are driest (data provided by NOAA ESRL Global Monitoring Division, Boulder, Colorado, USA; http://esrl.noaa.gov/). However, because of underlying volcanic geology, only 0.25 cm per year becomes stream flow within the Rio de Flag, making it one of the lowest percentage runoff systems in Arizona (Bills & Enyedy, 2015). Our ‘Central’ reach (~3 km long) is supported by the Rio de Flag Water Reclamation Plant in central Flagstaff. This plant began discharging into the river in 1993, was upgraded in 2009, and discharges approximately 2.4 million liters of tertiary treated effluent per day into the reach (City of Flagstaff, 2020; personal correspondence: Erin Young, City of Flagstaff). Our other Rio de Flag study reach, ‘Wildcat’ (~4 km long), is located approximately 10 km downstream of the Central reach, and is supported by the Wildcat Hill Water Reclamation Plant. This plant began discharging into the river in 1971, was upgraded in 2008, and discharges approximately 12 million liters of tertiary treated effluent per day into the reach (City of Flagstaff, 2020; personal correspondence: Erin Young, City of Flagstaff).

Figure 2 Photos showing examples of specific water quality sampling sites along the six study reaches.

The three reaches of the Santa Cruz River ((A) Tres Rios (32.435°, −111.233°), (B) Nogales (31.562°, −111.045°), and (C) Agua Nueva (32.287°, −111.030°)), the sole reach of the Salt River ((D) 91st Ave reach (33.381°, −112.322°)), and both reaches of the Rio de Flag ((E) Central (35.183°, −111.631°), and (F) Wildcat (35.234°, −111.543°)).

Data collection

To understand how water quality changes downstream from effluent outfalls, at least three sites were sampled for water quality within each of the six study reaches. The exact location and number of sampling sites for each reach were selected based on the availability of public access points and the length of reach, resulting in a range of three to eight sampling sites per reach (Table 1). A minimum of 0.8 km separated each progressive downstream sampling site within a reach, with an average distance of 3.5 km between sites (Fig. 1). To examine seasonal variation in water quality, we took measurements (described below) at each site at least once per season (winter = January to March, spring = April to June, summer = July to September, fall = October to December) in the year of 2018. To provide additional temporal resolution, one reach was sampled bi-monthly (Santa Cruz River at Nogales: 6x per year) and one reach was sampled monthly (Santa Cruz River at Tres Rios: 12x per year) (Table 1).

During each sampling event, we measured basic water quality parameters (dissolved oxygen, temperature, pH, specific conductance, and alkalinity) and nutrients (total phosphorus, ammonia, and nitrate). Specific conductance is hereafter referred to as conductivity. Basic water quality parameters were measured using in situ water quality probes: dissolved oxygen (Apera Instruments AI480 DO850 probe), pH, temperature, and conductivity (Apera Instruments SX823-B multiprobe). For nutrient concentrations (total phosphorus, total ammonia nitrogen, and nitrate), we collected 500 mL water samples from each site, immediately transported them to the laboratory at University of Arizona, and stored at 4 °C until analyzed using a YSI EcoSense Model 9500 Photometer (within 48 h). Total phosphorus was analyzed using phosphate LR method, then the phosphate was converted into total phosphorus; the ammonia test is based on an indophenol method; and finally the nitrate was analyzed using a nitratest method (YSI Inc., 2010). The intensity of the color produced in the test is proportional to those parameter concentrations and was quantitatively optically measured. The detection limits of total phosphorus, ammonia, and nitrate were 0.001, 0.01, and 0.2 mg/L, respectively. When nutrient analysis results exceeded the test’s calibration range, samples were diluted and rerun, and the appropriate dilution factor correction was applied.

Data analysis

Each of the eight water quality parameters were modeled with linear mixed effects models (LMEs: Pinheiro & Bates 2000) using distance from effluent outfall (Distance), Season, and the interaction between distance and season (Distance*Season) as predictors, with reach location (Site) as a random factor. Models were run in the statistical program R (version 3.5.1: R Core Team, 2019) with the package ‘glmmTMB’ (version 1.0.2.1; Brooks et al., 2017). Histogram analysis indicated that there were no outliers in the dataset and all variables were approximately normally distributed, except for ammonia, which was log-transformed prior to analysis (data not shown). Finally, to visualize how overall water quality varied across the year at all sites, we included all eight water quality parameters in a principal component analysis (PCA). The resulting plot allowed us to identify clustering of water quality conditions across river basins, reaches, and seasons. PCA was run with PC-ORD version 7.08 (McCune & Mefford, 2018).

Results

The effects of distance, season, and their interaction varied across specific water quality parameters and study reaches. Some water quality parameters, such as temperature, dissolved oxygen, and ammonia, showed pronounced changes with distance from outfalls across all reaches (Table 2), as we describe in detail below.

Table 2 Summary of Linear Mixed-Effect Model results explaining variation in temperature, dissolved oxygen, and ammonia.

The variation in temperature, dissolved oxygen, and ammonia with predictors distance from outfall in km (Distance), season (Spring, Summer, Winter), and the interaction between distance and season with river reach (Reach) as a random factor in the model. Results include Beta estimates (Estimates) with corresponding 95% confidence intervals (CI) and p values (p) as well as within reach residual variance (σ2), between reach variance (τ00), and intra-class correlation coefficient (ICC) for the random factor. Both marginal and conditional pseudo R2 values are listed. Bold indicates significant p values of predictors (α ≤ 0.05).

	Temperature	Dissolved oxygen	Ammonia	
Predictors	Estimates	CI	p	Estimates	CI	p	Estimates	CI	p	
(Intercept)	20.33	[17.57–23.09]	<0.001	8.66	[7.39–9.92]	<0.001	−0.65	[−0.99 to −0.32]	<0.001	
Distance	−0.14	[−0.23 to −0.06]	0.001	0.09	[0.04–0.15]	0.001	−0.02	[−0.04 to −0.01]	0.003	
Spring	4.97	[3.15–6.80]	<0.001	−1.89	[−3.08 to −0.64]	0.003	−0.16	[−0.52 to 0.19]	0.364	
Summer	7.70	[5.89–9.50]	<0.001	−3.63	[−4.83 to −2.42]	<0.001	0.15	[−0.20 to 0.49]	0.413	
Winter	−1.34	[−3.14 to 0.45]	0.143	−0.50	[−1.70 to 0.70]	0.418	−0.19	[−0.54 to 0.15]	0.274	
Distance*Spring	0.18	[0.06–0.30]	0.003	0.00	[−0.08 to 0.08]	0.945	0.00	[−0.02 to 0.02]	0.937	
Distance*Summer	0.15	[0.03–0.27]	0.016	0.03	[−0.05 to 0.11]	0.437	0.01	[−0.01 to 0.03]	0.348	
Distance*Winter	0.06	[−0.05 to 0.18]	0.262	−0.03	[−0.11 to 0.04]	0.377	−0.01	[−0.03 to 0.02]	0.593	
Random effects:										
σ2	7.37			3.29			0.28			
τ00 Reach	9.31			1.36			0.08			
ICC	0.56			0.29			0.22			
NSite	6			6			6			
N	170			170			170			
Marginal R2	0.550			0.349			0.227			
Conditional R2	0.801			0.540			0.397			

Water temperature was significantly affected by both distance from effluent outfall and season across all reaches (marginal R2 = 0.55, Table 2). Temperature values generally decreased with increasing distance from effluent outfalls (p ≤ 0.001), and values were usually higher in spring (mean: 27.2 °C; range: 19.8–34 °C) and summer (mean: 29.1 °C; range: 20.3–34.2 °C) and lower in fall (mean: 20.1 °C; range: 9.2–28.5 °C) and winter (mean: 19.2 °C; range: 14.1–25 °C). (Table 2; Fig. 3). Additionally, there was a significant modeled interaction between distance and season, where temperatures actually increased with distance from outfall in spring (p = 0.003) and summer (p = 0.016) in some reaches (Table 2, Fig. 3). However, this pattern of increasing temperatures with distance in warmer months was not observed in the higher elevation Nogales reach of the Santa Cruz River (Fig. 3B), highlighting some of the variability in factor responses across reaches. This variability across reaches was also evident in the increase in model explanatory power between the marginal and conditional R2 values (0.55 vs. 0.81, respectively; Table 2) when reach was included as a random factor.

Figure 3 Graph showing relationship between temperature (°C) and distance (km) from outfall.

(A) Tres Rios Reach and (B) Nogales reaches of the Santa Cruz River. Each point represents a sampling site, with points coded by season. To illustrate longitudinal trends, solid lines represent the best fit linear regression corresponding to each season.

Dissolved oxygen also varied significantly by both season and distance from outfall (marginal R2 = 0.349; Table 2). Concentrations of dissolved oxygen were lower during summer (mean: 5.9 mg/L; range 2.9–9 mg/L) and spring (mean: 7.3 mg/L; range: 2.7–14.3 mg/L) and higher during winter (mean: 8.4 mg/L; range: 5.4–15.6 mg/L) and fall (mean: 9.3 mg/L; range: 4.8–20.6 mg/L) (p < 0.003; Fig. 4). Furthermore, dissolved oxygen concentrations were consistently lowest at sampling sites closest to effluent outfalls, and then increased with distance downstream in each season (p = 0.001; Fig. 4). The lowest dissolved oxygen concentrations observed in our study were ~3 mg/L near the effluent outfall of the Tres Rios reach of the Santa Cruz River in summer (Fig. 4A).

Figure 4 Relationships between dissolved oxygen (mg/L) and distance (km) from outfalls.

(A) Tres Rios and (B) Nogales reaches of the Santa Cruz River. Each point represents a sampling site, with points coded by season. To illustrate longitudinal trends, solid lines represent the best fit linear regression corresponding to each season.

Ammonia concentrations decreased significantly with distance from effluent outfall in all reaches and seasons (p = 0.003), but there were no significant differences in ammonia concentrations by season (marginal R2 = 0.227; Table 2). The highest ammonia concentrations observed across all sites were at the Tres Rios reach of the Santa Cruz River, with values as high as 5.36 mg/L N during the fall season. However, these high ammonia concentrations decreased with the distance downstream and reached undetectable levels at the end of the reach, ~30 km downstream from the outfall (Fig. 5). In other seasons, ammonia concentrations remained very low or undetectable across all sampling sites.

Figure 5 Ammonia, nitrate, and phosphorus levels with increasing distances from the effluent outfall in the Tres Rios reach of the Santa Cruz River.

(A) Ammonia, (B) nitrate, and (C) phosphorus. Each point represents a sampling site, with points coded by season. Solid lines represent the best fit regression corresponding to each season.

In contrast to the above discussed water quality parameters, pH, conductivity, alkalinity, total phosphorus, and nitrate exhibited no significant relationships with distance from effluent outfall (all p > 0.1), and values only occasionally varied by season (see Table S1). Even when values for these five water quality parameters differed significantly in at least one season, overall model explanatory power was low (marginal R2 < 0.1 in most cases). Values for these five parameters were much more affected by reach identity, as seen in the dramatic increases in model explanatory power between marginal and conditional R2 values for these parameters and relatively high values for between reach variance (τ00 Reach; e.g., conductivity LME: marginal R2 = 0.006, conditional R2 = 0.929, τ00 Reach = 344,409.29: Table S1). For example, the Salt River at 91st Avenue reach had much higher initial conductivity values than the other five study reaches, and those values further increased by about 70% within 6 km of the effluent outfall (Fig. 6B).

Figure 6 Relationship between conductivity (µS/cm) and distance (km) from outfalls.

(A) The Wildcat Hill reach of the Rio de Flag, (B) the 91st Avenue reach of the Salt, and (C) the Agua Nueva and (D) Nogales reaches of the Santa Cruz River. Each point represents a sampling site, with points coded by season. Solid lines represent the best fit linear regression corresponding to each season.

PCA ordination revealed at least three noticeable clusters of data points when considering all water quality parameters together (Fig. 7A). First, data points from the short 91st Ave reach of the Salt River formed a scattered open cluster apart from all other points, with the vector highlighting increased conductivity as being the primary cause of this distinction (Fig. 7A). Second, observations from the long Tres Rios reach and medium-length Agua Nueva reach of the Santa Cruz River overlapped in the center of the plot, indicating no single water quality parameter was responsible for that cluster. Third, data points from the higher elevation Nogales reach of the Santa Cruz River and Central and Wildcat Hill reaches of the Rio de Flag formed a partly separate cluster that was associated with higher nitrate values. Data points did not form any visibly distinct clusters by season. However, a few patterns were evident from examining the vectors. Observations from the summer tended to have higher temperatures and ammonia levels, and lower pH levels, than those from other seasons (Fig. 7B).

Figure 7 PCA ordinations of all eight water quality factors across all six samples reaches and seasons.

The same ordination is coded here by (A) reach and (B) season to illustrate clusters. The loadings of the eight water quality factors on the two primary axes are visualized with the blue vectors.

Discussion

Distance effect

The effect of longitudinal distance from effluent outfalls on water quality varied widely among measured parameters. Effluent is generally warmer than natural baseflow temperatures, so heat exchange between the atmosphere and stream likely led to longitudinal cooling trends (Boyle & Fraleigh, 2003), especially during winter (Beganskas & Toran, 2021; also see “4.2 Seasonal effects” below). As temperatures cooled, dissolved oxygen levels rose. Increased oxygen levels were likely due to a combination of cooler temperatures, photosynthesis by macrophytes and algae, and atmospheric reaeration (O’Connor, 1967; Correa-González et al., 2014).

In contrast, longitudinal variation in pH was not significant and values remained circumneutral to slightly basic. These results are in line with other studies of streams receiving effluent, which reported relatively stable pH values downstream of effluent outfalls (Chen et al., 2009; Prat et al., 2013; Matamoros & Rodríguez, 2017). Relatedly, we also found no longitudinal patterns in alkalinity, which indicates the buffering capacity to neutralize acids and bases of a water body. Some previous studies have reported a decrease in alkalinity with distance downstream (Birge et al., 1989; Boyle & Fraleigh, 2003), a result likely due to dilution by other water sources. However, our effluent-dependent study reaches are not fed by any perennial tributaries that can dilute the effluent during baseflow.

Ammonia levels decreased significantly with increasing distances from effluent outfalls. High levels of ammonia have commonly been reported near effluent outfalls, with decreasing levels downstream (Sebenik, Cluff & DeCook, 1972; Hamdhani, Eppehimer & Bogan, 2020). Although trends across reaches were equivocal, in the long Tres Rios reach of the Santa Cruz River, the average rate of ammonia loss over ~31 km in the fall season was about 10 times greater than nitrate gain (−0.11 mg/L/km of N ammonia; 0.01 mg/L/km of N nitrate). In summer, the average rate of ammonia loss was double that of nitrate gain (−0.08 mg/L/km of N ammonia; 0.04 mg/L/km of N nitrate) (Fig. 5). We presume the seasonal differences in nutrient gain and loss are likely due to temperature. Thus, nitrification occurring in the stream channel (Sebenik, Cluff & DeCook, 1972) does not fully account for ammonia removal, suggesting mechanisms besides oxidation of ammonia to nitrate contributed to the observed pattern. These other mechanisms could include volatilization, adsorption by cation exchange complex, fixation by clays, and fixation by organic matter (Van Schreven, 1968; Weiler, 1979; Freney, Simpson & Denmead, 1981; Simon & Kennedy, 1987; Kowalchuk et al., 1998; Lin et al., 2019). Since pH was always <9 across all reaches and seasons (see Supplemental Material), total ammonia nitrogen was present mainly as ammonium ion (NH4+), thus volatilization is not expected to have contributed significantly to ammonia loss in any of the study reaches.

In contrast to our findings with ammonia, total phosphorus and nitrate did not change significantly with increasing distance from effluent outfalls. Previous studies in the other effluent-fed streams (Birge et al., 1989; Chen et al., 2009) indicated phosphorus concentration decreased with distance downstream. This reduction is likely due to biotic and abiotic nutrient uptake along the reaches (Davis & Minshall, 1999; Stutter, Demars & Langan, 2010). When we look at our longest reaches, there appears to be a trend toward decreasing total phosphorus, but a lack of response over shorter distances in other reaches may have precluded the global LME model from detecting an effect of distance.

In natural streams, conductivity tends to be low in headwater reaches and then increase in downstream reaches, especially when streams pass through urban areas (Chusov, Bondarenko & Andrianova, 2014). However, previous studies of effluent-fed streams have reported elevated conductivities near outfalls and then relatively stable values downstream (Chen et al., 2009; Prat et al., 2013; Matamoros & Rodríguez, 2017). In this study, we did not observe a significant effect of distance from effluent outfall on conductivity; instead values tended to be stable. However, patterns in one reach were noticeably different. The Salt River reach had both higher baseline levels of conductivity and also a significant increase in values moving downstream from the effluent outfall (Fig. 6). These patterns could be due to urban inputs (i.e., non-point sources) that were not mapped as part of our research, or due to underlying soil conditions. For example, the Salt River Valley has many areas that are high in soluble mineral salt (Harper, 1931).

Seasonal effects

Seasonal atmospheric and climatic factors, such as air temperature, solar radiation, relative humidity, cloud cover, and wind speed, work together to shape water temperatures through heat exchange between the atmosphere and the stream (Sinokrot & Stefan, 1994; Beganskas & Toran, 2021). This effect is usually stronger in unregulated stream systems (Bowles, Fread & Greeney, 1977) and relatively weak in effluent-dependent stream systems, especially near effluent outfalls (Hamdhani, Eppehimer & Bogan, 2020). However, we did observe a significant effect of season on water temperatures across all six study reaches, with warmer water observed in spring and summer than in fall and winter, regardless of reach length (Table 2; Fig. 3). This pattern demonstrates the strong influence of atmospheric and climatic conditions on water temperature despite the stable, warm temperature of effluent at its point of release (Beganskas & Toran, 2021). Trends in dissolved oxygen levels across seasons were generally opposite to those of temperature, with the lowest values observed in summer; however, this pattern was only significant in three of the six reaches (Table 2; Fig. 4). In the other three reaches, dissolved oxygen levels might be more strongly driven by non-temperature factors, such as variation in biochemical oxygen demand, water volume, and photosynthetic activity (Guasch & Subater, 1995; Mulholland, Houser & Maloney, 2005; Mandal, Upadhyay & Hasan, 2010).

Nutrient levels varied across seasons but not in a predictable way across reaches (Table 2), which may be related to seasonal variation in raw wastewater influent and performance of each wastewater treatment plant. The most substantial seasonal variation in nutrient levels occurred on the Tres Rios reach of the Santa Cruz River, which was also our longest study reach (Fig. 5). In summer and fall, ammonia levels were much higher and nitrate levels were at their lowest, relative to concentrations observed in winter and spring, especially nearer to the effluent outfall. This finding was unexpected because conversion of ammonia to nitrate during wastewater and potable water treatment is usually highest in warmer seasons (Siripong & Rittmann, 2007; Liu et al., 2017). We suspect that manipulation in treatment operations at the Tres Rios WRF contributed to lessened nitrification capacity in summer (Hegg, Rakness & Schultz, 1979; Liu et al., 2012). Finally, phosphorus levels in effluent from the Tres Rios WRF were higher during summer and spring, and lower during winter and fall. We speculate that this may be related to the type of Accumulibacter involved in phosphorus removal processes during wastewater treatment. Some types of Accumulibacter alter their activity patterns with seasonal-induced temperature changes (Flowers, Cadkin & McMahon, 2013).

Conductivity and pH differed significantly by season across five of the six study reaches, but we could not detect any consistent patterns (Table 2; Fig. 5). For example, conductivity was often highest in the summer, but in some cases it was lowest in the summer (Fig. 5), and similar inconsistencies were observed with pH (Supplemental Materials). Both factors can be influenced by precipitation and runoff, especially in urban areas where runoff may carry pollutants (Vega et al., 1998; Laudon, Westling & Bishop, 2000). However, we avoided sampling during periods of precipitation and subsequent runoff, thus all water quality measurements reflect conditions when flow was 100% effluent. Further study is needed to understand why these factors vary inconsistently by season in the six study reaches.

Geography and climate

Geographic and climatic settings strongly influence the structural and functional features of natural rivers and streams (Gasith & Resh, 1999; Shi et al., 2019), but it is unclear how strongly effluent-dependent systems might be influenced. For example, we expected that water temperature would be fairly consistent across all reaches, regardless of climate zone, given the warm temperature of effluent resulting from the treatment process (Brooks, Riley & Taylor, 2006; Hamdhani, Eppehimer & Bogan, 2020). However, when considering all measured factors across all reaches, sampling sites, and sampling dates together, distinct clusters were observed for high (>1,000 m), medium (~700 m), and low elevation sites (<300 m) (Fig. 7). As previously mentioned, the lowest elevation reach (91st Ave, Salt River) occurs in a stream basin with natural salt deposits (Harper, 1931). The high mineral content in the water likely contributed to the distinctness of that reach (Fig. 7A). Higher nitrate levels also were observed in the higher elevation reaches. Because rates of biochemical reactions, including denitrification, are temperature-dependent (Dawson & Murphy, 1972; Xu, Dai & Chai, 2019), lower air temperatures may reduce nitrate removal capacity of the treatment plants at higher altitudes.

In contrast to the distinct geographic clusters in the PCA ordination, we observed broad overlap of water quality conditions by season (Fig. 7B). This lack of pattern is likely due to idiosyncratic seasonal changes across the factors and reaches, as described in the previous section and observed in previous studies (Hamdhani, Eppehimer & Bogan, 2020). Thus, although individual reaches may exhibit clear seasonal trends in water quality, the variation in the significance of those trends across reaches in different locations obscures detection of any overall seasonal patterns (Gardner & McGlynn, 2009; Pratt & Chang, 2012).

Potential for effluent to support native aquatic species

The ability of effluent to enhance or re-create habitat for native aquatic species is still actively debated, but some conservation successes have been noted in recent years (Halaburka et al., 2013; Luthy et al., 2015). In our six study reaches, effluent has restored flow to rivers that had been dry for decades due to extraction of groundwater and surface water (e.g., Webb et al., 2014). However, prior to wastewater treatment plant upgrades, water quality was generally poor and aquatic biodiversity was low (Cordy et al., 2000; Walker, Goforth & Rector, 2005; Sonoran Institute, 2017). In particular, high water temperatures, low dissolved oxygen levels, and elevated concentrations of ammonia have been cited as the cause of low aquatic biodiversity in effluent-dependent streams (e.g., Monda, Galat & Finger, 1995; Hamdhani, Eppehimer & Bogan, 2020; Eppehimer et al., 2020).

Many aquatic animals are adapted to specific temperature ranges (Carveth, Widmer & Bonar, 2006; Eliason et al., 2011; Nikel et al., 2021) and warmer water can affect their growth, behavior and survival (Crawshaw, 1977; Schneider & Connors, 1982; Marine & Cech, 2004). Among 11 native Arizona fish species tested by Carveth, Widmer & Bonar (2006), the most sensitive to high water temperatures was the Speckled dace (Rhinichthy osculus), which becomes disoriented at 34 °C and perishes at 36 °C. One of the most tolerant fishes identified was the Gila topminnow (Poeciliopsis occidentalis), which can withstand temperatures as high as 38–39 °C for short periods of time (Carveth, Widmer & Bonar, 2006). The highest temperature we recorded across all of our sites and seasons was 34.2 °C, but not all instantaneous measurements were taken at the hottest times of the day for each reach, so it is possible that temperatures reach 1 °C to 3 °C higher (Lowney, 2000). These findings suggest that effluent-dependent in Arizona are generally thermally suitable for native fish, but sensitive species could be negatively impacted in the warmest reaches. In these cases, a gradient in fish composition could occur, where thermally sensitive species are only found further downstream from the outfalls. A similar pattern has been reported for the thermally-sensitive Greenside darter (Etheostoma blennioides) in a Canadian effluent-fed stream (Brown et al., 2011).

Dissolved oxygen concentration also can be a direct indicator of the ability of a waterbody to support aquatic life. We occasionally observed lower dissolved oxygen levels near effluent outfalls in summer, with concentrations as low as 2.9 mg/L, as has been observed in other effluent-fed systems (Birge et al., 1989; Boyle & Fraleigh, 2003; Matamoros & Rodríguez, 2017). Mortality or loss of equilibrium for fishes and other aquatic organisms can occur at concentrations between 1 and 3 mg/L (US EPA, 1986), and chronic exposure can cause behavioral changes that make individuals vulnerable to predation or other risk factors (Dean & Richardson, 1999). For native Arizona fishes, dissolved oxygen levels ranging from 0.22 to 1.47 mg/L have been reported as being lethal (Lowe, Hinds & Halpern, 1967). These findings suggest that oxygen levels near effluent outfalls might be stressful for native fishes during warmer seasons.

Aquatic invertebrates can also be affected by low oxygen levels, with tolerant worms (Oligochaeta: Martins, Stephan & Alves, 2008) and fly larvae (e.g., Chironomus, Chironomidae: Lencioni et al., 2008) replacing sensitive mayflies (Ephemeroptera) and stoneflies (Plecoptera) near effluent outfalls (Hamdhani, Eppehimer & Bogan, 2020). In fact, a previous study from the Santa Cruz and Salt Rivers reported low oxygen levels and only a few tolerant invertebrate taxa near effluent outfalls (Cordy et al., 2000). However, wastewater treatment plants supplying these reaches have since been upgraded, and recent findings from the Santa Cruz River demonstrate a robust aquatic invertebrate community with few apparent dissolved oxygen limitations (Eppehimer et al., 2020). Although invertebrate studies are not available for the other two rivers (Salt and Rio de Flag), dissolved oxygen measurements suggest that diverse aquatic invertebrate communities could be found there too.

Ammonia has direct toxic effects on aquatic species (e.g., Richardson, 1997; Hickey & Vickers, 1994), with concentrations greater than 2 mg/L as N (pH 7.0 and temperature 20 °C) causing impairment of aquatic life (Constable et al., 2003; Yeom et al., 2007; US EPA, 2013). The vast majority of our measurements in Arizona were below this threshold, but at one site on the Santa Cruz River we did find concentrations as high as 3-5 mg/L in summer and fall. In a prior study of the Santa Cruz River, before treatment plants were upgraded, the absence of sensitive mayfly taxa was likely a result of elevated ammonia concentrations (Boyle & Fraleigh, 2003). However, recent work has shown that both mayflies and fish are now found near effluent outfalls that experience episodic, but not chronic, high ammonia levels (e.g., lower Santa Cruz River: Sonoran Institute, 2017; Eppehimer et al., 2020). Together, these findings suggest that ammonia is no longer a primary concern for aquatic species in effluent-dependent streams of Arizona.

We suggest that in order to improve water quality for native aquatic species, wastewater treatment plants should only release tertiary treated effluent. Furthermore, designing minimum discharge goals to maintain flow for at least several kilometers is recommended (Eppehimer et al., 2020). These longer lengths of flow allow additional natural remediation of tertiary treated effluent to occur, as shown in our improvements to dissolved oxygen and ammonia, for example. Low dissolved oxygen near effluent outfalls can also be improved through installation of physical structures that promote aeration of the water (Alp & Melching, 2011). Additionally, stable flow with significant volume will allow riparian gallery forests to develop, the shade from which helps moderate water temperatures and increase dissolved oxygen levels during summer months (Ghermandi et al., 2009; Dugdale et al., 2018). Finally, diurnal flow fluctuations are common in effluent dependent river systems, which can be problematic for aquatic biota (Eppehimer et al., 2021). Retention or detention structures can be engineered to help slow the flow of the water and reduce the magnitude of diurnal effluent fluctuations in recipient streams.

Conclusion

Distance from effluent outfalls, season, and climatic and geographic factors all play important roles in the water quality dynamics of effluent-dependent streams in Arizona. Water quality conditions deteriorated somewhat in some reaches during the hottest months of the summer and, for several factors, we observed natural remediation of water quality in longer reaches. Our study reaches met or exceeded water quality conditions needed to support robust assemblages of native species. However, conditions may be stressful for the most sensitive taxa at sites closest to effluent outfalls, especially in summer. Natural streams across arid and semi-arid regions are continuing to dry up due to climate change and water abstraction (Seager et al., 2007; de Graaf et al., 2019), but effluent-dependent streams are becoming more common (Luthy et al., 2015; Hamdhani, Eppehimer & Bogan, 2020). Our findings suggest that these systems have the capacity to serve as refuges for native biota, and they may become the only aquatic habitat available in many urbanizing arid regions. This is especially important to document given the competing demands for effluent as water resources in arid and semi-arid regions become more scarce (Wolfand et al., 2022). Quantifying the ecological value of effluent-dependent streams will be essential to ensuring they continue to flow and support native aquatic species into the future.

Supplemental Information

Supplemental Information 1 Raw data of water quality measurement.

Click here for additional data file.

Supplemental Information 2 Summary Linear Mixed-Effect model results explaining variation in alkalinity, conductivity, pH, nitrate, and phosphorous (total phosphorous).

With predictors distance from outfall in km (Distance), season (Spring, Summer, Winter), and the interaction between distance and season with river reach (Reach) as a random factor in the model. Results include Beta estimates (Estimates) with corresponding 95% confidence intervals (CI) and p values (p) as well as within-reach residual variance (σ2), between reach variance (τ00), and intra-class correlation coefficient (ICC) for the random factor. Both marginal and conditional pseudo R2 values are listed. Bold indicates significant p values of predictors (α ≤ 0.05).

Click here for additional data file.

Supplemental Information 3 Flow discharge volume, maximum and minimum air temperatures on all sampling dates for each site and reach across the study period.

Click here for additional data file.

Supplemental Information 4 Average, minimum, and maximum water quality factor values across all sites by season.

Click here for additional data file.

Supplemental Information 5 Locations of all sampling sites within each of the six study reaches across the three focal river basins in Arizona.

Click here for additional data file.

Supplemental Information 6 Water quality graphs of all sampling sites.

Click here for additional data file.

We thank K. Hollien for field and laboratory work, E. Young for discharge data for the Rio de Flag, as well as E. McGee, S. Wasko, and M. Grageda for providing useful feedback.

Additional Information and Declarations

Competing Interests

Author Contributions

Data Availability

The authors declare that they have no competing interests.

Hamdhani Hamdhani conceived and designed the experiments, performed the experiments, analyzed the data, prepared figures and/or tables, authored or reviewed drafts of the article, and approved the final draft.

Drew E. Eppehimer conceived and designed the experiments, performed the experiments, analyzed the data, prepared figures and/or tables, authored or reviewed drafts of the article, and approved the final draft.

David M. Quanrud conceived and designed the experiments, authored or reviewed drafts of the article, and approved the final draft.

Michael T. Bogan conceived and designed the experiments, performed the experiments, analyzed the data, prepared figures and/or tables, authored or reviewed drafts of the article, and approved the final draft.

The following information was supplied regarding data availability:

The raw data is available in the Supplemental Files.

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
