# Peer review of "Seasonal and longitudinal water quality dynamics in three effluent-dependent rivers in Arizona"

_PeerJ, doi:10.7717/peerj.15069_

## Round 0.1 · original submission · Major Revisions

I have received feedback from three reviewers who suggest that your manuscript requires revision. I invited you to revise your manuscript by giving due consideration to the comments and suggestions from reviewers.

·

Basic reporting

The article was written well in English and the structure of the article is acceptable for publication in Peer J. Introduction section needs to get improvement. The abstract should be rewritten with adding conclusive results in numbers. Literature and references related to the methods used in this study should be added to introduction section. Even though introduction includes the study importance, the model and statistical method used in the study also have to be mentioned with details.

Experimental design

This paper is enought novelty and explain the gap related in the literature. This submission was investigated the seasonal dynamics of basic water quality parameters in six effluent-dependent reaches of differing 80 lengths in three river systems across a climate gradient in Arizona (USA). Even though introduction includes the study importance, the model and statistical method used in the study also have discussed with integrated result.Please give more information regarding the methods used to analyze nutrients. The discussion section should be improved. Some paragraphs are far from discussing the results and only gives background informations

Validity of the findings

The abstract needs improvement to better describe the aim of the study and major findings. Please revise discussion section, but there is background information where need to be moved to introduction section. In addition, this section should include an integrated interpretation of the results of each method used in the study. The authors should include suggestions for increasing water quality at the study area.

In conclusion section lines 423-424: The sentence “Distance from effluent outfalls, season, and climatic and geographic factors all play important roles in the water quality dynamics of effluent-dependent streams” should be revised and relevant revisions should be made in the text. In "Conclusions" please specify the contribution of this study to the relevant literature.

Reviewer 2 ·

Basic reporting

The analysis in the Abstract is qualitative primarily analysis, and the quantitative analysis results are lacking

Subsection 2.1 Fig 1 must be improved to understand better the geographical setting (e.g the international reader who is unfamiliar with the USA must understand where on the map the study was conducted).

Experimental design

No comment

Validity of the findings

Line 185-188 I suggest the authors elaborate on how each parameter is based on the effect of distance as this term “showed consistent effects of distance from outfalls” is a be skeletal. It gives the impression that the reading was the same throughout which is not the case when looking at table 2.
Line 202 -205 please give the specific or range of the readings for dissolved oxygen during summer, spring, winter and fall. Then this will be supplemented by the level of significance as shown in the manuscript.
Line 208-210 Please show the different readings of ammonia concentration (quantitative) in the different seasons with distance from the effluent outfall. This will later be followed by “Ammonia concentrations decreased significantly with distance from effluent outfall in all reaches and seasons (p = 0.003), but there were no significant differences in ammonia concentrations by season (marginal R2 = 0.227; Table 2)”.
Line 225- 227 “For example, the Salt River at 91st Avenue reach had much higher initial conductivity values than the other five study reaches, and those values further increased by about 70% within 6 km of the effluent outfall (Fig. 6).” Not clear as figure 6 has four graphs from A to D therefore, it should be shown clear which graph e.g. Fig 6 B

Additional comments

Main comment
The research has been well organized and conducted.
Recommend to the author (s) of the research to briefly mention about the situation of groundwater in the area such as whether it is used for irrigation and drinking water purposes in the area or not. If it is used for irrigation, there might be some depletion of it is level in dry periods because of limited recharge from rainfall. Thus, it also has some negative impact on the quality of groundwater in a long period. It should be better to give brief information regarding that if there is available data about that.

Annotated reviews are not available for download in order to protect the identity of reviewers who chose to remain anonymous.

Reviewer 3 ·

Basic reporting

no comment

Experimental design

no comment

Validity of the findings

no comments

Additional comments

The article is entitled 'Seasonal and longitudinal water quality dynamics in three effluent-dependent rivers in Arizona'. The article is interesting, but in my opinion it needs to be improved:
- The research area is poorly described. The map with the location of the research area needs to be improved. The map should include the hydrographic network, catchment boundaries, water sampling sites, and sewage treatment plants. The map should contain information on the topography and geological structure, as well as on land use.
- The authors did not characterize the climatic conditions that prevailed in the research period, a graph with air temperature and precipitation is needed, which concerns the background, which is the basis for further water quality research. You need a table with the measurement dates and the corresponding weather conditions.
- The article lacks information on the size of the flows in the rivers. The flow is important when analyzing the influence of pollutants on water quality, it is important to know about the flow increment - Figure 3 shows the relationship between temperature and distance. It should be clearly indicated whether it is the temperature of the water or the air temperature



The reference needs to be supplemented with new items, issued after 2020

---

## Round 0.2 · accepted · Accept

I am pleased to inform you that your revised manuscript has been accepted for publication.

·

Basic reporting

no comment

Experimental design

no comment

Validity of the findings

no comment

Additional comments

The article meets the PeerJ criteria and should be accepted as is.

Reviewer 2 ·

Basic reporting

No comment

Experimental design

No comment

Validity of the findings

No comment

Additional comments

No comment

Reviewer 3 ·

Basic reporting

no comment

Experimental design

no comment

Validity of the findings

no comment

Additional comments

The authors have corrected the article.